# A Comprehensive Review of Natural Products as Therapeutic or Chemopreventive Agents against Head and Neck Squamous Cell Carcinoma Cells Using Preclinical Models

**DOI:** 10.3390/biomedicines11092359

**Published:** 2023-08-23

**Authors:** Yoon Xuan Liew, Lee Peng Karen-Ng, Vui King Vincent-Chong

**Affiliations:** 1Oral Cancer Research & Coordinating Centre (OCRCC), Faculty of Dentistry, University of Malaya, Kuala Lumpur 50603, Malaysia; yoonxuan1996@gmail.com; 2Department of Oral Oncology, Roswell Park Comprehensive Cancer Center, Buffalo, NY 14263, USA

**Keywords:** head and neck squamous cell carcinoma, natural products, phytochemicals, chemotherapeutics, chemoprevention

## Abstract

Head and neck squamous cell carcinoma (HNSCC) is a type of cancer that arises from the epithelium lining of the oral cavity, hypopharynx, oropharynx, and larynx. Despite the advancement of current treatments, including surgery, chemotherapy, and radiotherapy, the overall survival rate of patients afflicted with HNSCC remains poor. The reasons for these poor outcomes are due to late diagnoses and patient-acquired resistance to treatment. Natural products have been extensively explored as a safer and more acceptable alternative therapy to the current treatments, with numerous studies displaying their potential against HNSCC. This review highlights preclinical studies in the past 5 years involving natural products against HNSCC and explores the signaling pathways altered by these products. This review also addresses challenges and future directions of natural products as chemotherapeutic and chemoprevention agents against HNSCC.

## 1. Introduction

Head and neck cancer (HNC) represents cancers occurring in the head and neck region, which includes the lip and oral cavity, nasal cavity, larynx, and pharynx [1]. More than 90% of HNCs are head and neck squamous cell carcinomas (HNSCCs), which account for approximately 300,000 deaths and 500,000 new cases worldwide annually [2]. In the United States, HNSCC has been diagnosed as one of the top 10 leading cancers in men in 2022 [2,3,4]. The low survival rate of HNSCC has been proposed to be associated with cancer recurrence, distant metastases, the progression of second primary cancers, and resistance to chemo/radiotherapy [5,6], making it a public health issue that compromises patients’ quality of life.

Tobacco smoking, excessive alcohol drinking, betel quid chewing, and high-risk human papillomavirus (HPV) have been documented as risk factors for HNSCC [7,8,9,10]. Clinical intervention of HNSCC often takes place at advanced stages of the disease due to late diagnoses, especially among individuals with a lower socioeconomic background [11,12]. The most common clinical interventions for HNSCC are surgery, radiotherapy, chemotherapy, or combined therapy, which causes numerous side effects during the treatment of this disease [11]. However, even with a successful clinical intervention, approximately 30% of these patients treated at advanced stages of the disease develop recurrent locoregional or second primary cancers, with the onset of chemo- or radio-resistance [13,14,15]. Notably, these clinical interventions are only effective on a limited subgroup of HNSCC patients and often result in additional morbidities.

Cetuximab, an epidermal growth factor receptor (EGFR)-targeting monoclonal antibody, was the first molecular-targeted drug approved by the U.S. Food and Drug Administration (FDA) as a chemotherapeutic agent for HNSCC in 2006 [5,16]. A retrospective study in Japan reported a total effective rate of 57.1%, a median progression-free survival (PFS) of 5.5 months, and an overall survival (OS) of 8.0 months with cetuximab in locally advanced HNSCC, while a total effective rate of 60.0%, a PFS of 3.8 months, and an OS of 5.8 months were reported in distantly metastatic HNSCC [17]. However, the chemoresistance ability of certain mutated cancer cell types, such as EGFRvIII, has shown resistance toward cetuximab [18]. In recent years, both pembrolizumab and nivolumab, which act as anti-programmed cell death receptor 1 (PD-1) immunotherapy drugs, were approved by the FDA for recurrent or metastatic HNSCC treatment [19,20]. However, most patient previously exposed to anti-PD-1 monoclonal antibody develop acquired resistance to immunotherapeutic drugs, making it difficult to treat recurrent or metastatic cancers [21]. There is certainly an urgent need for alternative therapeutic agents to overcome the acquired resistance of HNSCC to the standard of care. Because of the challenges faced using immunotherapies, there is a great need for therapeutic agents that can work effectively as chemoprevention or to enhance the effectiveness of chemotherapeutic agents when used in combination to kill cancer cells.

Natural products are compounds naturally found in natural resources such as plants, which possess biological activities [22]. In recent years, natural products have been widely reported for their chemotherapeutic and chemoprevention properties against HNSCC due to their low cytotoxicity, efficacy against cancers, availability, and low cost [12,22]. According to U.S. National Cancer Institute, chemoprevention is defined as the use of certain drugs or other substances to help lower a person’s risk of developing cancer or keep it from coming back [23]. The beneficial properties of natural products are consistent and could overcome the challenges with current treatment such as acquired chemoresistance, cytotoxicity against normal cells, and expensive therapy. Numerous studies have been carried out with various natural products on preclinical models of HNSCC, including various HNSCC cell lines and xenograft or carcinogen-induced tumor animal models (Figure 1). For example, psorachromene, a flavonoid found in Psoralea corylifolia, which has been used in traditional Chinese medicine (TCM) and Ayurveda, had shown therapeutic effects against HNSCC via regulation of the EGFR signaling pathways and other carcinogenesis-related signaling pathway, making it a strong candidate to act as a chemotherapeutic agent [24]. Other natural products, such as calcitriol, have been reported to possess chemoprevention properties against carcinogen-induced HNSCC in animal models [25]. Therefore, natural products have shown potential as candidates for further exploration as adjuvant, neoadjuvant chemotherapy, and chemoprevention agents in HNSCC. However, we acknowledge that, in many cases, these compounds are studied only at the beginning state in animal models. More studies on safety and efficacy are needed to improve the therapeutic potential in patients, which include clinical trials.

The activation of complex signaling pathways, such as phosphoinositide 3-kinase/protein kinase B/mammalian target of rapamycin (PI3K/Akt/mTOR) and mitogen-activated protein kinase/extracellular signal-regulated protein kinase (MAPK/ERK), is important for tumor development, cell survival, and angiogenesis, while epithelial to mesenchymal transition (EMT) signaling leads to tumor invasion and migration [26,27]. Thus, the inhibition of these signaling pathways could lead to cell death, inhibiting tumor development and metastasis [27]. Notwithstanding the extensive knowledge at the molecular level of tumor-associated signaling pathways, the chemotherapeutic agents targeting the oncogenic pathways are limited [28]. To date, Crooker et al. and Rahman et al. have described the use of natural products as chemoprevention agents against HNSCC [29,30]. Several natural products, such as vitamin A, green tea extract, and curcumin, which have shown promising results in preclinical studies, were reported to show toxicity with limited bioavailability in clinical studies [31,32,33,34]. Vitamin A has been shown by Shin et al. [31] and Papadimitrakopoulou et al. [33] to induce toxicity via oral mucosa and lip inflammation, conjunctivitis, skin reactions, fatigue, and joint and muscle pain. On the other hand, poor oral absorption of green tea extract and curcumin has been observed, which limits the bioavailability of both natural products [33,34]. Other emerging chemotherapeutic phytochemicals or herbal derivatives against HNC have recently been described by Aggarwal et al. [12]. However, several new phytochemicals against HNSCC, such as actein, calcitriol, and psorachromene, were not fully addressed. Therefore, in this review, the comprehensive mechanisms of various natural products showing significant preclinical results in HNSCC models for the past 5 years are discussed.

## 2. Chemotherapeutic Properties of Natural Products on Essential Pathways for HNSCC

### 2.1. PI3K/Akt/mTOR Pathway

It was previously reported that the activation of the PI3K/Akt/mTOR pathway has been observed in approximately 90% of HNSCC cases, making it a prominent target for treatments of HNSCC [35,36]. The activation of PI3K/Akt/mTOR signaling also plays an important role in HNSCC chemotherapy and radiotherapy resistances, as the inhibition of the signaling pathway has shown positive effects on tumor proliferation and radiotherapy sensitization in preclinical studies [37,38]. The PI3K/Akt/mTOR pathway is activated when a ligand-like growth factor binds with the receptor tyrosine kinase (RTK), leading to the activation of PI3K, which, in turn, partially activates Akt [39]. Then, mTORC2 is required to completely activate Akt via phosphorylation, leading to the activation of multiple proteins involved in cell proliferation and motility [40]. Inhibition of Akt formation will therefore limit the expression of oncoprotein. Various natural products, namely actein, salicylate, tanshinone IIA, xanthohumol, fucoidan, honokiol, ilimaquinone, nimbolide, and cinnamaldehyde, have led to the downregulation of Akt expression and phosphorylation of Akt (p-Akt) using HNSCC preclinical models [41,42,43,44,45,46,47,48,49]. The downregulation of mTOR expression in HNSCC preclinical models has been shown by salicylate, honokiol, cinnamaldehyde, and *Seco-A*-ring oleanane, with PI3K also limiting the phosphorylation of Akt [42,45,48,50]. The expression of glycogen synthase kinase-3β (GSK-3β) plays an important role in regulating growth, cell cycle progression, apoptosis, and cancer cell invasion in HNSCC [51,52]. The inactivation of GSK-3β via phosphorylation has previously shown significant inhibition in cancer cell growth and migration in HNSCC [51]. Preclinical studies with nimbolide have also shown the upregulation of phosphorylated GSK-3β, inactivating GSK-3β and thus inhibiting cell proliferation [47]. The expression of c-Myc has been associated with the upregulation of Akt-related pathways, leading to cancer cell proliferation and tumorigenesis [53]. The downregulation of c-Myc using HNSCC preclinical models has been observed with tanshinone IIA, leading to the inhibition of tumorigenesis [43].

### 2.2. MAPK/ERK Pathway

Activation of the MAPK/ERK signaling pathway, which includes signaling molecules such as interleukin-8 (IL-8), vascular endothelial growth factor (VEGF), mitogen-activated extracellular protein kinase (MEK), and extracellular signal-regulated protein kinase (ERK), is strongly associated with the expression of oncoproteins leading to cell proliferation and angiogenesis [54]. Honokiol, sodium Danshensu, cinnamaldehyde, and protocatechuic acid have been reported to inhibit MAPK/ERK signaling in HNSCC preclinical models, thus inhibiting cell proliferation [45,48,55,56]. Protocatechuic acid inhibits MAPK/ERK signaling via activating the c-Jun N-terminal kinase/p38 (JNK/p38) signaling pathway [56]. However, the expression of JNK plays a dual role in HNSCC, both tumor-suppressive and -progressive, due to the complex crosstalk between multiple signaling molecules and pathways [57]. Therefore, understanding of the role of JNK and p38 in HNSCC is required to ensure relevant clinical research in future. The overexpression of matrix metalloproteinases (MMPs), such as MMP-2 and MMP-9, has been associated with the progression and metastasis of HNSCC [58]. Fucoidan, sodium Danshensu, [(3β)-3-hydroxy-lup-20(29)-en-28-oic acid], and Seco-A-ring oleanane have been reported to downregulate MMP-2 [49,50,55]. Meanwhile, fucoidan and trichodermin downregulate MMP-9 in HNSCC preclinical models [49,56], inhibiting cancer cell metastasis. The expression of VEGF plays an important role in blood vessel formation, known as angiogenesis, which, in turn, promotes cancer cell growth due to the supply of nutrients via newly formed blood vessels [59,60]. It was also noticed that VEGF could be upregulated by human growth factor (HGF) through the PI3K/Akt/mTOR and MAPK signaling pathways [61]. Preclinical studies of HNSCC have indicated that cinnamaldehyde, [(3β)-3-hydroxy-lup-20(29)-en-28-oic acid], and Seco-A-ring oleanane could downregulate VEGF, thus inhibiting angiogenesis [48,50].

### 2.3. NF-κB and STAT3 Transcription Factors

Nuclear factor-κappa B (NF-κB), a protein complex consisting of five transcription factors, namely RelA, RelB, c-Rel, NF-κB1, and NF-κB2, has been reported to play an important role in cell proliferation and survival in HNSCC [62,63]. Preclinical studies using HNSCC models have demonstrated the inhibition of NF-κB, leading to apoptosis with neferine, trichodermin, cinnamaldehyde, and *Seco-A*-ring oleanane [48,50,64,65].

The expression of the signal transducer and activator of transcription-3 (STAT3), a transcription factor from the STAT family, plays an important role in cell proliferation, survival, and metastasis in HNSCC [66,67,68]. Phosphorylation of STAT3 (p-STAT3) has been proposed as a crucial reaction for the complete activation of STAT3, which is regulated by p38, EGFR, and janus kinases (JAKs) [69,70]. The activation of STAT3 would also lead to various STAT3-dependent pathways, such as the interleukin-8/STAT3 (IL-8/STAT3) and EGFR/STAT3/SRY-box transcription factor 2 (EGFR/STAT3/SOX2) pathways, which have been shown to play important roles in cancer stemness [71,72]. Preclinical studies in HNSCC models have indicated the downregulation of p-STAT3 with trichodermin and *Seco-A*-ring oleanane [50,65]. Meanwhile, psorachromene caused the downregulation of EGFR, leading to apoptosis and inhibiting metastasis [24].

Actein, a biologically active compound found in the rhizome of *Cinicifuga foetida*, has been discovered to inhibit forkhead box O1 (FoxO1) by upregulating the phosphorylation of FoxO1 (p-FoxO1), leading to the inhibition of cell proliferation via the Akt/FoxO1 signaling pathway [41]. Knockdown of FoxO1 would reverse the antiproliferation properties of actein, indicating that the Akt/FoxO1 pathway plays an important role in actein-induced effects toward HNSCC [41]. FoxO1 is one of many transcription factors among the FoxO family which regulates various biological activities in HNSCC, such as cancer cell invasion and proliferation [73].

Activated p53 via phosphorylation acts as a tumor suppressive molecule, which has been associated with induced apoptosis in HNSCC [74,75]. About half of HNSCC cases have shown a loss-of-function p53 gene mutation [76], making p53 an interesting target in improving the efficiency of HNSCC therapy through the reactivation of p53. In HPV-positive HNSCC, p53 was downregulated by the E6 oncoprotein, leading to tumorigenesis [76]. Notably, neferine and ilimaquinone have been shown to upregulate or activate p53 in preclinical HNSCC studies, leading to apoptosis [46,64].

### 2.4. LC3-Dependent Autophagy

Autophagy is a major process involving protein degradation followed by the turnover of cellular components, which helps maintain the intercellular homeostasis [77]. Light chain 3 (LC3) has been strongly associated with the regulation of autophagosome, a double-membrane vesicle formed during autophagy [77,78]. During the formation of an autophagosome, cytosolic LC3 (LC3-I) is converted into the activated form of intra-autophagosomal LC3 (LC3-II) via a uibiquitylation-like reaction catalyzed by Atg7 and Atg3, by conjugating with phosphatidylethanolamine [77,78,79]. Therefore, the increased conversion of LC3-I to LC3-II indicates higher levels of autophagosome and thus the activation of autophagy, leading to tumor suppression [80]. The accumulation of p62 was also found to be associated with the induction of autophagy in HNSCC [81]. Honokiol, ilimaquinone, and nimbolide have been reported to increase LC3-II/LC3-I ratio in HNSCC preclinical models [42,45,47].

Paradoxically, autophagy may potentially lead to tumor survival when certain conditions are met [82,83,84,85]. Hypoxia, a condition where the oxygen level is below the physiological level, is a common feature in tumor progression when the oxygen supply does not meet the demand due to the exponential growth of tumor [83,86]. Hypoxia-induced autophagy in tumors has been observed to induce tumor survival in vitro [83]. Meanwhile, the knockdown of essential autophagy proteins with a xenograft model was found to promote tumor suppression [84]. The turnover of cellular components by autophagy may essentially help with hypoxia and nutrient stress faced by exponentially growing tumors, thus enhancing tumor survival [83,84]. 

### 2.5. Bcl-2/Bax Signaling

Bcl-2 and Bax are proteins found in the mitochondrial membrane, nuclear envelop, and endoplasmic reticulum [87,88]. Bcl-2 inhibits the release of cytochrome c, which impedes tointrinsic apoptosis, while Bax reverses the reaction, leading to the induction of intrinsic apoptosis in HNSCC [89]. Cytochrome c, when released followed by a cascade of reaction involving apoptotic protease activating factor 1 (Apaf-1), Caspase-9, Caspase-7, and Caspase-3, causes apoptosis [89,90]. A preclinical study with fisetin showed that the upregulation of cytochrome c leads to apoptosis [91]. Preclinical studies in HNSCC with actein, xanthohumol, fucoidan, ilimaquinone, nimbolide, 4-O-methylhonokiol, cinnamaldehyde, kaempferol, and fisetin have indicated higher a Bax/Bcl-2 ratio (upregulating Bax and/or downregulating Bcl-2) on the mitochondrial membrane, supporting the induction of intrinsic apoptosis of HNSCC [41,44,46,47,48,49,91,92]. The activation of Caspase-8 has been associated with the activation of Caspase-7 and Caspase-3 followed by apoptosis induction, while Survivin acts as an anti-apoptotic protein [75,93,94]. In preclinical studies of HNSCC, Ilimaquinone was reported to upregulate Caspase-8, while the downregulation of Survivin has been observed with actein, xanthohumol, ilimaquinone, and seco-A-ring oleanane [41,44,46,50].

### 2.6. Cell Cycle Arrest by Cyclin and the CDK Signaling Pathway

The cell cycle acts as a fundamental process for cancer progression in HNSCC, including cell proliferation and differentiation [95]. It is well established that cyclins and cyclin-dependent kinases (Cdks) play an important role in the regulation and transition of cell cycles [96,97,98]. G0/G1 transitioning is strongly dependent on cyclin-D/Cdk4/6 complexes; S phase entry is dependent on the cyclin E/Cdk2 complex; S/G2 transitioning is dependent on the cyclin A/Cdk2 complex, followed by mitotic phase entry, which is dependent on the cyclin A/Cdk1 complex, and finally, M/G0 transitioning, which is dependent on the cyclin B/Cdk1 complex [99,100,101,102,103]. The inhibition of the cell cycle via the formation of cyclin-Cdk complexes would lead to cell cycle arrest and, eventually, cell death in HNSCC [104]. Anti-mitogenic signals such as p16, p21, and p53 could inhibit the cell cycle transition effectively by inhibiting the formation of cyclin-Cdk complexes, thus acting as an important target for cell cycle arrest induction [99,105,106]. Xanthohumol, neferine, fucoidan, hydroxygenkwanin, ilimaquinone, honokiol, trichodermin, cinnamaldehyde, resveratrol, and curcumin have been discovered to induce cell cycle arrest via cyclin-dependent signaling with HNSCC preclinical models [44,45,46,48,49,56,64,107,108].

### 2.7. Potential Natural Products as Therapeutic Agent for HNSCC

Actein, salicylate, honokiol, trichodermin, psorachromene, protocatechuic acid, fucoidan, hydroxygenkwanin, nimbolide, 4-O-methylhonokiol, [(3β)-3-hydroxy-lup-20(29)-en-28-oic acid], seco-A-ring oleanane, cinnamaldehyde, kaempferol, resveratrol, and curcumin induced cell cycle arrest via the regulation of cyclins and Cdks, which play an important role in coordinating the cell cycle progression [24,41,42,45,47,48,49,50,56,65,91,92,107,108]. The expressions of p21, p27, p16, and p53, acting as cyclin-Cdk complex inhibitors, have also been upregulated by actein, hydroxygenkwanin, honokiol, ilimaquinone, tanshinone IIA, resveratrol, and curcumin, which leads to induced cell cycle arrest [41,43,45,46,107,108].

Cell migration and angiogenesis have been widely studied and strongly associated with regulations of MMPs. Fucoidan, trichodermin, [(3β)-3-hydroxy-lup-20(29)-en-28-oic acid], seco-A-ring oleanane, and sodium danshensu have shown to downregulate MMPs, which inhibit cell migration and angiogenesis in HNSCC [49,50,55,65]. The VEGF and VEGF receptors play an important role in coordinating angiogenesis, while ilimaquinone, cinnamaldehyde, [(3β)-3-hydroxy-lup-20(29)-en-28-oic acid], and seco-A-ring oleanane have shown inhibition of the VEGF signaling in vitro [46,48,50].

The PI3K/Akt/mTOR and MAPK/ERK signaling pathways have been well studied and have shown a strong association with the cell proliferation and survival of HNSCC. In vitro studies with actein, salicylate, tanshinone IIA, fucoidan, honokiol, xanthohumol, ilimaquinone, cinnamaldehyde, nimbolide, and seco-A-ring oleanane have disrupted the PI3K/Akt/mTOR signaling pathway, leading to the inhibition of cell proliferation and survival [41,42,43,44,45,46,47,48,49,50]. In contrary, in vitro studies with honokiol, sodium danshensu, protocatechuic acid, and cinnamaldehyde have shown inhibition of the MAPK/ERK signaling pathway, leading to the inhibition of cell proliferation and survival [45,48,55,56].

Cell apoptosis via the mitochondrial pathway is regulated by caspases, where the proteases are activated by the Bax and Bcl-2 proteins present in the mitochondrial membrane. In vitro studies have shown that the downregulation of Bcl-2 and upregulation of Bax by fucoidan, ilimaquinone, nimbolide, 4-O-methylhonokiol, actein, and cinnamaldehyde leads to a cascade reaction of apoptosis induction [41,46,47,48,49,92]. The upregulation of caspases leading to apoptosis have also been shown by in vitro studies with hydroxygenkwanin, ilimaquinone, nimbolide, psorachromene, actein, conocarpan, trichodermin, protocatechuic acid, kaempferol, and fisetin [24,41,46,47,56,65,91,107,109]. Table 1 summarizes in vitro preclinical studies using natural products, while Table 2 summarizes in vitro and in vivo preclinical studies of natural products.

## 3. Chemoprevention Properties of Natural Products against HNSCC Oral Carcinogenesis Mechanism

Oral carcinogenesis often involves the formation of abnormalities in the oral tissue, known as oral potentially malignant disorder (OPMD), before proceeding to oral squamous cell carcinoma (OSCC), a major type of HNSCC [111,112]. Previous studies have found the malignant transformation (MT) rate of OPMD to be 7.9%, while high-risk OPMD, such as erythroplakia, has shown an average MT rate of 33.1% [113]. To date, no preventive strategies including the use of drugs and/or surgical procedures has been considered the standard of care for OPMDs, thus indicating the need to investigate the chemoprevention properties of various natural products against oral carcinogenesis [114]. Several drugs, such as celecoxib, erlotinib, and metformin, have been investigated for their HNSCC prevention properties via clinical trials on oral premalignant lesions [115,116,117]. However, the use of erlotinib and celecoxib has shown no significant result in reducing the oral cancer-free survival rate while possessing higher toxicity [115,116]. The use of metformin has shown a low clinical response rate (17%) in terms of reduction in lesion size [117].

In vivo studies involving the use of genetically altered rodents or rodents treated with chemical carcinogens could lead to site-specific carcinogenesis, mimicking carcinogenesis in humans [118]. Moreover, 4-nitroquinoline 1-oxide (4NQO) acts as a tobacco-mimicking carcinogen, which has been widely used in carcinogen-induced HNSCC animal models, mainly due to the similarity in terms of genetic alteration and expression between 4NQO-induced mouse models and human oral carcinogenesis [119,120]. Other carcinogens, including 7,12-dimethylbenz(a)anthracene (DMBA) and dibenzo[a,l]pyrene (DBP), have also been widely used as HNSCC-inducing agents in animal models [64,121].

Several review studies have introduced various natural products as chemoprevention agents against HNSCC, where vitamin A, green tea extracts, and curcumin have shown promising results in preclinical and clinical trials [29,30]. However, vitamin A has shown toxicity [31,32], while green tea extracts and curcumin [33,34] have both shown limitations in bioavailability. In a randomized chemoprevention trial reported by Papadimitrakopoulou et al. [32], low-dose 13-cis retinoic acid (a derivative of vitamin A) could induce grade 1 (45%), 2 (37%), 3 (15%), and 4 (1%) toxicity, including cheilitis, conjunctivitis, and skin reactions. Similarly, a phase II chemoprevention trial by Shin et al. [31] with combinations of interferon-alpha, 13-cis retinoic acid, and alpha-tocopherol induced mild to moderate non-hematologic toxicity. One patient was reported with a severe throat infection due to beta-hemolytic streptococci which required an emergency tracheostomy. The patient was still able to complete the planned treatment after fully recovering from the infection [31]. Green tea extract was reported to induce adverse effects such as insomnia, nausea, nervousness, and headache, which is most likely due to the presence of caffeine in green tea extract [33]. The poor oral absorption of epigallocatechin-3-gallate, the most abundant polyphenol in green tea extract, was reported by Tsao et al. [33], leading to variability in plasma epigallocatechin-3-gallate concentrations. Similarly, a phase I chemoprevention trial by Cheng et al. [34] with curcumin also indicated the poor gastrointestinal absorption of curcumin, as the peak serum curcumin concentration was recorded at 1.77 µM with 8000 mg daily dosage. The poor bioavailability of both green tea extract and curcumin have introduced difficulties in dosage estimation as absorption varies among patients, which may lead to ineffective treatment. Therefore, in this review, we seek to provide a greater variety of promising natural products with chemoprevention properties using 4NQO, DMBA, or DBP-induced carcinogenesis animal models [25,64,121]. Table 3 summarizes preclinical studies with natural products involved in the chemoprevention of HNSCC investigated in the past 5 years.

Preclinical in vivo studies involving neferine and nimbolide have indicated an inhibition in carcinogenesis and a reduction in the tumor growth of DMBA-induced HNSCC, making both possible chemoprevention agents [41]. Similarly, calcitriol successfully inhibited 4NQO-induced HNSCC carcinogenesis [25]. Interestingly, a study by Vincent-Chong et al. [25] with calcitriol on a 4NQO-induced animal model showed the influence of treatment stage intervention and duration of exposure to treatment in carcinogenesis. Understanding the pathways involved during carcinogenesis will allow effective treatment intervention in HNSCC patients.

Calcitriol, nimbolide, and neferine are being investigated in preclinical studies for chemoprevention, and these products have shown promising results by reducing or inhibiting carcinogenesis-related molecular mechanisms [25,47,64].

## 4. Limitation and Future Direction

As multiple signaling and crosstalk between pathways occur during carcinogenesis, with most recurrent or metastatic HNSCC failing the primary standard of care, an effective treatment for cancer may require a combined therapeutic approach such as the use of multiple signaling inhibitors combined with DNA-damaging drugs for the most efficient outcome [26]. Therefore, the synergic or antagonistic effects of natural products with standards of care such as chemotherapy (cisplatin, cetuximab, pembrolizumab and nivolumab) and radiotherapy should be analyzed using preclinical models. However, fewer than 10% of the reviewed studies reported the combination effects of natural products with the standard of care. For instance, xanthohumol, psorachromene, honokiol, calcitriol, and salicylate were shown to provide synergistic effects with standard-of-care treatment such as chemotherapy and radiotherapy using HNSCC preclinical models [24,25,42,44,45].

The major reason for the low survival rate of HNSCC is due to late diagnoses and risk factors associated with HNSCC progression, which lead to the risk of recurrent or metastatic SCC [5,12]. Chemoprevention therapy could potentially act as an important barrier to lower the risk of recurrent or metastatic SCC and the malignant transformation of OPMD; therefore, chemoprevention should be widely studied in the future. The prevention of HNSCC involving single-agent chemotherapy, such as retinoids and isotretinoin, possesses high toxicity and low efficacy, indicating the need for the development of new chemoprevention agents either as alternative or adjunctive agents for HNSCC prevention. In the current review, only 5 out of 37 studies explored the potential usage of natural products as chemoprevention therapy using 4NQO/DMBA/DBP-induced oral carcinogenesis [25,47,64,121]. Furthermore, the stage and duration of natural products’ intervention on oral carcinogenesis should also be extensively explored given the [25] strong association with the progression of carcinogenesis. All six natural products (Table 3) are strongly encouraged to proceed with clinical trials for high-risk OPMD patients, as previous trials with celecoxib, erlotinib, and metformin on mild to advanced OPMD showed no significant clinical improvements [115,116,117].

A preclinical study reported by Dai et al. [122] using various cancer lines (melanoma, breast, colon, and liver cancer cell lines (H1299, BT549, MDA-MB-231, MDA-MB-468, SW620, MHCC97H, and B16F10)) indicated the crucial role of immunomodulatory roles of natural products in anti-cancer treatments, including the involvement of CD3+ CD8+ T lymphocytes in a co-culture system together with cancer cell lines. Similarly, Cattanaeo et al. [123] and Neal et al. [124] have proposed the use of co-culture organoid-tumor reactive T lymphocyte system to investigate the role of anti-PD-1 drugs on T lymphocyte activity. Other studies have also used similar systems to screen natural product-derived drugs and epigenetic inhibitors for the non-cytotoxic T lymphocyte immunomodulating effects [122,125]. Apart from the in vitro co-culture system proposed, Verma et al. [126] implemented an in vivo RPMOC1 synergic HNSCC animal model to investigate the effects of a non-oncological drug, calcitriol, on the immunomodulation of T lymphocytes, which was the only preclinical study involving synergic HNSCC in an in vivo model present at the time of preparing this review. The initiation of this review was prompted by a recent publication by Crooker et al. [29] which provided informaion on the use of natural products in HNSCC preclinical models in 2018. Over the past five years, there have been limited efforts to publish reviews related to natural products, with one notable publication by Aggarwal et al. [12] focusing solely on HPV-related HNSCC. To address the paucity of information, we explored the possibilities of using the latest research of natural products with in vitro and in vivo studies, listed in Table 1, Table 2 and Table 3, that has not been discussed in previous review papers, particularly targeting HNSCC, as some signaling pathways are common among different subtypes. The review also highlights various challenges in the field that hinder the clinical translation of natural products.

Among the reviewed natural products, actein, salicylate, tanshinone IIA, xanthohumol, honokiol, trichodermin, psorachromene, and protocatechuic acid have been investigated as chemotherapeutic agents with HNSCC cell lines and xenografted animal models, and these products have shown promising targeted molecular mechanisms against HNSCC, making them ideal candidates for further clinical trials for safety and efficacy analyses [24,41,42,43,44,45,56,65]. Furthermore, salicylate and xanthohumol have been investigated to provide synergistic effect with cisplatin and radiotherapy, respectively, making them the strongest candidates for future clinical trials for HNSCC or OPMD patients [42,44]. Finally, calcitriol, nimbolide, and neferine were the major natural products investigated in this review; these products showed promising chemoprevention properties against induced-carcinogenesis animal models. We encourage further investigation into the safety and efficacy of these products with human trials, as well as in high-risk OPMD such as erythroplakia and leukoplakia patients [25,47,64]. Nevertheless, it is crucial that the results of preclinical studies presented in this manuscript serve only as a principle for further investigations and should not be directly extrapolated into clinical practices without additional detailed evaluation. Preclinical studies, either in vitro or in vivo, provide valuable insights on the potential of natural products as therapeutic agents; however, their efficacy, toxicity profile, and applicability in human patients should be widely explored. Finally, caution must be exercised when interpreting the findings of preclinical studies and their potential implication for clinical trials.

## 5. Conclusions

Both chemotherapeutic and chemoprevention approaches showed promising anti-cancer effects through preclinical studies involving various HNSCC cell lines and animal models via various pathways (Figure 2). However, the lack of extensive molecular mechanisms of natural products and lack of combination of natural products with the current standard of care limits the use of natural products as new chemotherapeutic drugs. The only study that used calcitriol to determine the immunomodulatory effect of natural product in the context of HNSCC highlighted the effort to investigate the role of these natural products since the immune checkpoint inhibitor has been recognized as one of the standards of care for HNSCC patients. The low intrinsic toxicity of natural products in normal cells and significant therapeutic effects toward cancer cells have sparked an interest in oncology studies in recent years. The synergistic effect of natural products with standard of cares, including chemotherapy, radiotherapy, and immunotherapy, has showed promising properties as both alternative and adjunctive chemotherapeutic or chemoprevention agents in cancer treatment or prevention, respectively.

As a conclusion, we attempted to provide a comprehensive database of natural products used for in vivo and in vitro preclinical studies involving HNSCC in the recent years, which would facilitate the identification of effective natural products that show promising chemotherapeutic and chemoprevention properties against HNSCC while possessing low toxicity. Natural products have shown to be promising molecular targets for therapy and prevention against HNSCC, making them great alternatives and adjunctive agents in cancer treatment. However, further toxicity profiles should be analyzed for natural products in future clinical trials to ensure the safety and efficacy of natural products against HNSCC.

## Figures and Tables

**Figure 1 biomedicines-11-02359-f001:**
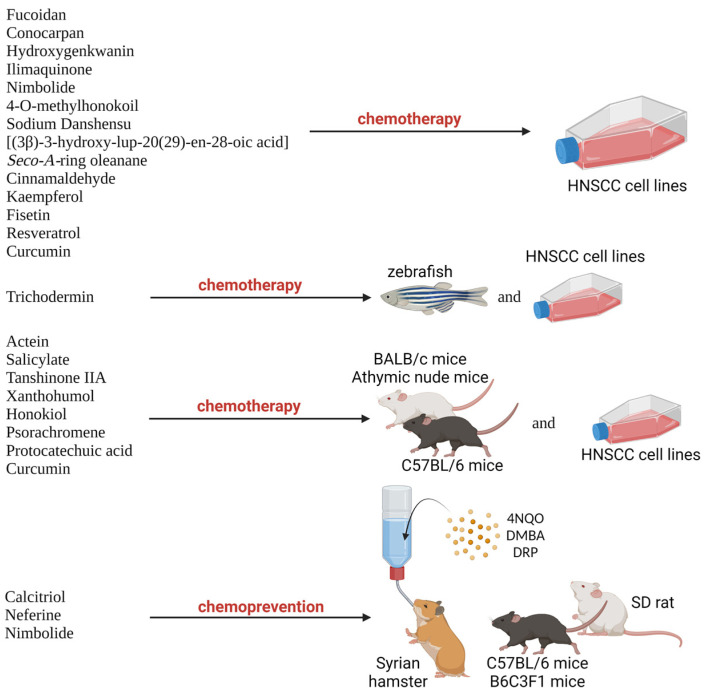
Natural products involved in preclinical trials involving HNSCC cell lines and animal models.

**Figure 2 biomedicines-11-02359-f002:**
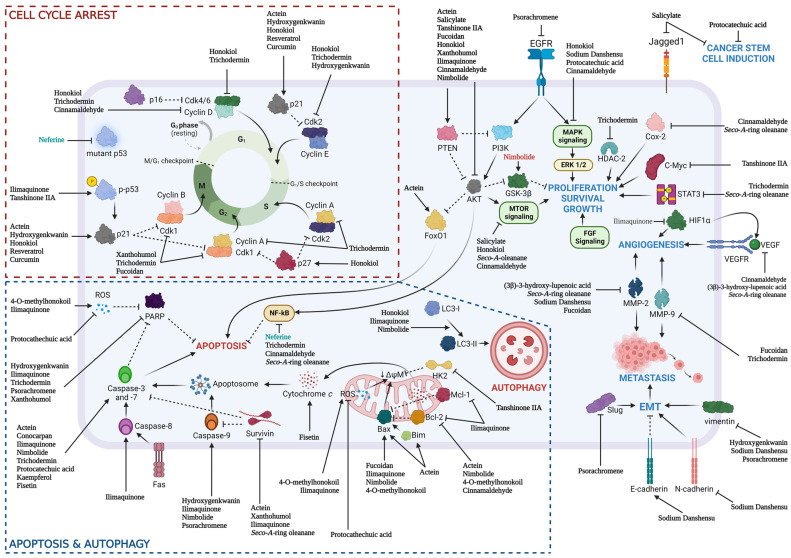
Various natural products and their respective molecular mechanisms against HNSCC: chemoprevention (natural product name in green), chemotherapeutic (natural product name in black).

**Table 1 biomedicines-11-02359-t001:** Chemotherapeutic findings from preclinical studies with natural products involving in vitro HNSCC cell lines.

No	Author	PMID	Natural Product	Preclinical Model (Cell Lines)	Test & Dosage	Findings	Pathway Related
1	Zhang et al. [49]	31495936	Fucoidan	In vitro (SCC15, SCC25)	Cell Proliferation; MTT assay (IC_50_ value; 2 µM, 5 µM) Cell Cycle Arrest; flow cytometry (dosage not available) Colony Formation; clonogenic assay (dosage not available) Apoptosis; Annexin V assay (dosage not available) Cell Migration and Invasion; transwell assay, wound healing assay (dosage not available)	Reduced cell viability and inducing cell arrest (G_2_ Phase) Inducing intrinsic apoptosis via upregulation of Bcl-2 associated X protein (Bax) expression Downregulating phosphorylation of protein kinase B (p-Akt) and cyclin-dependent kinase 1 (Cdk1) Inhibiting cell migration and invasion via downregulation of metalloproteinases (MMP-9) and MMP-2 expression Upregulate circRNA filament A (circFLNA) expression, mediating signaling molecules	Akt Bax/Bcl-2
2	Fonseca et al. [109]	33078660	Conocarpan	In vitro (SCC9, SCC4, SCC25)	Cell Viability; MTT assay (1.2–300 µM) Apoptosis; cell morphology observation	Reduced cell viability and promote cell apoptosis Activating Caspase-3 expression and inducing pyknotic nuclei	Caspase
3	Huang et al. [107]	31620368	Hydroxygenkwanin	In vitro (SAS & OECM1)	Cell Proliferation; MTT assay (25, 50 & 75 µM) Cell Cycle Arrest; flow cytometry (25, 50 & 75 µM) Colony formation; clonogenic assay (25, 50 & 75 µM) Cell Migration and Invasion; wound healing assay and invasion assay (25 & 50 µM)	Reduced cell growth and colony formation Activated p21 and inhibited Cdk2 expression, inducing cell cycle arrest Upregulating poly (ADP-ribose) polymerase (PARP) cleavage and phosphorylated X-linked inhibitor of apoptosis protein (p-H2AX) Induce cell apoptosis via intrinsic pathway involving Caspase-9 Inhibition of cell invasion and migration via downregulation of Vimentin	Caspase
4	Lin et al. [46]	32825464	Ilimaquinone	In vitro (SCC4, SCC2095)	Cell Viability; MTT assay (IC_50_ value; 7.5 µM, 8.5 µM) Apoptosis; Annexin V assay (5, 10, 20 & 30 µM) Autophagy; autophagic vesicle detection (2.5, 5, 10 & 20 µM)	Reduced cell viability Upregulated caspase-3, caspase-8, caspase-9 and PARP cleavage leading to apoptosis Upregulated proapoptotic protein Bax and p-p53, and downregulated anti-apoptotic protein myeloid leukemia cell differentiation protein (Mcl-1), B-cell lymphoma 2 (Bcl-2), and apoptotic inhibitor protein Survivin Downregulated p-Akt and hypoxia-inducible factor 1-alpha (H1F-1α), mediating cell migration Upregulated p-H2AX, as regulation due to increased reactive oxygen species (ROS) generation Induced autophagy by upregulating light chain 3 (LC3-II) and autophagy related 5 (Atg5) expression	Akt Caspase Bax/Bcl-2
5	Sophia et al. [47]	30352996	Nimbolide	In vitro (SCC131, SCC4)	Cell Viability; MTT assay (IC_50_ value; 6 µM, 6.2 µM) Cell Cycle Arrest; flow cytometry (6 µM, 6.2 µM) Apoptosis; Annexin V, nuclear morphology, mitochondrial transmembrane potential (6 µM, 6.2 µM) Autophagy; autophagic vesicle detection (6 µM, 6.2 µM)	Reduced cell viability Upregulated Bax and downregulated Bcl-2, leading to intrinsic apoptosis Upregulated cleaved Caspase-9 and Caspase-3 expression Induced conversion of LC3-I to LC3-II, inducing autophagy Downregulating p-Akt and upregulation of phosphorylated glycogen synthase kinase 3 beta (p-GSK-3β), inhibiting phosphoinositide 3-kinase/protein kinase B/glycogen synthase kinase 3 beta (PI3K/Akt/GSK-3β)	PI3K/Akt/GSK-3β Caspase Bax/Bcl-2
6	Xiao et al. [92]	29332355	4-O-methylhonokoil	In vitro (PE/CA-PJ41)	Cell Viability; MTT assay (IC_50_ value; 2.5 µM) Cell Cycle Arrest; flow cytometry (1, 2.5 & 5 µM)	Induced G_2_/M cell cycle arrest and apoptosis Upregulated formation of intracellular ROS, leading to ROS-mediated reduction in mitochondrial membrane potential, inducing intrinsic apoptosis Upregulated Bax and downregulated Bcl-2 expression, leading to apoptosis	Bax/Bcl-2
7	Kumar et al. [55]	33101201	Sodium Danshensu	In vitro (FaDu, CA9-22)	Cell Viability; MTT assay (50 µM) Cell Migration and Invasion; wound healing, migration, and invasion assay (25, 50 & 100 µM)	Reduced cell motility, migration and invasion Upregulation of E-cadherin and zonula occludens-1 (ZO-1) expression, and downregulation of MMP-2, Vimentin and N-cadherin expression, leading to anti-migratory and anti-invasive effect Downregulate p38 phosphorylation leading to inhibition of mitogen-activated protein kinase (MAPK) signaling pathway and downregulation of extracellular signal-regulated protein kinase (ERK1/2)	MAPK/ERK
8	Aswathy et al. [50]	33860206	[(3β)-3-hydroxy-lup-20(29)-en-28-oic acid]	In vitro (SAS)	Cell Proliferation; MTT assay (IC_50_ value; 6 µM) Colony Formation; colony-forming assay (10 & 15 µM) Cell Cycle Arrest; flow cytometry (5, 10 & 20 µM) Apoptosis; Annexin V (50 µM) Cell Migration; cell migration assay (2.5 & 5 µM)	Reduced colony formation and migration and induced apoptosis Downregulated vascular endothelial growth factor (VEGF) and MMP-2 expression, via Akt/mTOR pathway	Akt/mTOR JAK/STAT3 VEGF NF-κB
			*Seco-A*-ring oleanane	In vitro (SAS)	Cell Proliferation; MTT assay (IC_50_ value; 20 µM) Colony Formation; colony-forming assay (10 & 15 µM) Cell Cycle Arrest; flow cytometry (1, 3 & 5 µM) Apoptosis; Annexin V (50 µM) Cell Migration; cell migration assay (2.5 & 5 µM)	Reduced colony formation and migration and induced apoptosis Downregulated cyclooxygenase-2 (Cox-2), Survivin, MMP-2 and VEGF, via nuclear factor kappa light chain enhancer of activated B cells (NF-κB), mTOR and STAT3 pathway	
9	Aggarwal et al. [48]	35774603	Cinnamaldehyde	In vitro (SCC9, SCC25)	Cell Viability; MTT assay (IC_50_ value; 40 µM, 45 µM) Colony formation; clonogenic assay (40 µM, 45 µM) Cell Cycle Arrest; flow cytometry (40 µM, 45 µM) Apoptosis; Annexin V (40 µM, 45 µM) Cell Invasion; matrigel cell invasion assay (40 µM, 45 µM)	Reduced cell viability and inhibited proliferation, migration and invasion Induced cell cycle arrest at G_2_/M and S-phase Induces autophagy Inhibited nuclear translocation of NF-κB from cytoplasm Cinnamaldehyde shows binding affinity with MAPK-p38α and dihydrofolate reductase (DHFR) Downregulating expression of NF-κB/p65, Cox-2, p110a, cyclin-D1, VEGF, Akt, mTOR, p-mTOR, and Bcl-2, and upregulating beclin-1 expression	PI3K/Akt/mTOR NF-κB MAPK
10	Kubina et al. [91]	37371038	Kaempferol & Fisetin	In vitro (SCC9, SCC25)	Cell Proliferation; WST-1 assay (IC_50_ value kaempferol; 45.03 µM, 49.90 µM & fisetin; 38.85 µM, 62.34 µM) Cell Cycle Arrest; flow cytometry (1/2 and ¼ IC_50_ value) Apoptosis; Annexin V Detection Kit (1/2 and ¼ IC_50_ value) Cell Migration; wound healing assay (1/2 and ¼ IC_50_ value)	Inhibited cell proliferation and migration Induced apoptosis by activation of Caspase-3 and decreased potential of mitochondrial membrane Downregulation of Bcl-2 Fisetin upregulate cytochrome c Kaempferol induced cell cycle arrest at S phase	Bax/Bcl-2
11	Bostan et al. [108]	32859062	Resveratrol & Curcumin	In vitro (PE/CA-PJ49)	Cell Viability; MTT assay (IC_50_ value resveratrol; 46.8 µM & curcumin; 16.3 µM) Cell Proliferation; cell proliferation assay (resveratrol; 40 µM & curcumin; 15 µM) Cell Cycle Arrest; flow cytometry (resveratrol; 40 µM & curcumin; 15 µM) Apoptosis; Annexin V Detection Kit (resveratrol; 40 µM & curcumin; 15 µM)	Inhibited cell proliferation Induced apoptosis Amplifying effect of low concentration of cisplatin on inhibition of cell proliferation, and induction of apoptosis and cell cycle arrest Upregulation of p21	Caspase

**Table 2 biomedicines-11-02359-t002:** Chemotherapeutic findings from preclinical studies with natural products involving in vitro HNSCC cell lines and in vivo xenograft models.

No	Author	PMID	Natural Product	Preclinical Model	Test & Dosage	Findings	Pathway Related
1	Zhao et al. [41]	34175854	Actein	In vitro (CAL27, SCC9)	Cell Proliferation; CCK-8 Assay (IC_50_ value; 15 µM, 12 µM) Cell Cycle Arrest; flow cytometry (7, 15 & 30 µM) Apoptosis; Annexin V assay (7, 15 & 30 µM)	Reduced cell viability and induced cell cycle arrest Induced apoptosis via upregulated Bax and downregulated Bcl-2 Downregulated Survivin and upregulated p21 and Bim Downregulating Akt and upregulating forkhead box protein O1 (FoxO1), inhibiting Akt/FoxO1 pathway	Akt/FoxO1 Bax/Bcl-2
				In vivo (CAL27) C57BL/6 mice	Mammary fat pad subcutaneous inoculation Intragastrically administration of 10, 20, and 50 mg/kg dosage	Impaired tumor growth	
2	Zhang et al. [42]	32267053	Salicylate	In vitro (SAS)	Cell Proliferation; MTS Assay (2500, 5000 and 10,000 µM) Cell Cycle Arrest; flow cytometry (5000 µM) Tumorsphere Formation; tumorsphere formation assay (5000 µM)	Combination with cisplatin enhanced the cytotoxicity and induced cell apoptosis Downregulated p-Akt, mTOR, p-S6 and p-70S6, inhibiting Akt/mTOR signaling pathway Downregulated Jagged-1, SRY-box transcription factor 9 (SOX9), yes-associated protein-1 (YAP-1), sonic hedgehog protein (Shh), aldehyde dehydrogenase (ALDH), oxtamer-binding transcription factor 4 (OCT4) expression	Akt/mTOR
				In vivo (SAS) Nude mice	Subcutaneous inoculation Oral administration of 3 mg/kg dosage	Monotherapy impaired tumor growth Combination with cisplatin enhanced tumor growth inhibition	
3	Li et al. [43]	32424132	Tanshinone IIA	In vitro (CAL27, SCC9, SCC15, SCC25)	Cell Viability; MTS assay (2 & 5 µM)	Reducing cell viability Inducing intrinsic apoptosis via downregulation of p-Akt, c-Myc and hexokinase 2 (HK2) Inhibiting glycolysis of SCC by downregulating expression of HK2 Promoting FBW7 E3 ligase interaction with c-Myc, shortening half-life of c-Myc	Akt/c-Myc
				In vivo (CAL27, SCC15) Athymic nude mice	Right flank subcutaneous inoculation Intraperitoneal administration of 10 mg/kg dosage	Reducing population of Ki-67 positive cells Downregulating p-Akt, c-Myc and HK2 Impaired tumor growth	
4	Li et al. [44]	32410646	Xanthohumol	In vitro (CAL27, SCC9, SCC15, SCC25)	Cell Viability; MTS assay (1, 2 & 5 µM)	Reducing cell viability and colony formation Regulating Akt/Wee1/Cdk1 signaling pathway Upregulating Bax on mitochondria, PARP cleavage and Caspase-3, leading to intrinsic apoptosis Promote ubiquitination and degradation of Survivin by upregulating FBXL7 E3 protein	Akt/Wee1/Cdk1 Bax/Bcl-2 Caspase
				In vivo (CAL27, SCC25) Athymic nude mice	Right flank subcutaneous inoculation Intraperitoneal administration of 10 mg/kg dosage	Delayed tumor development and impaired tumor growth Reducing population of Ki-67 positive cells Downregulating p-Akt and Survivin Sensitizing radioresistance cells to radiotherapy	
5	Huang et al. [45]	29363886	Honokiol	In vitro (OC2, OCSL)	Cell Viability; CCK-8 assay (IC_50_ value; 35 µM, 22 µM) Cell Cycle Arrest; flow cytometry (25 & 40 µM, 15 & 30 µM) Apoptosis; Annexin V assay (25 µM, 15 µM)	Reduced cell growth Induced cell cycle arrest at G_0_/G_1_ phase via upregulating p21 and p27, accumulation of cyclin-E, and downregulating Cdk2, Cdk4 and cyclin-D1 Inhibiting MAPK pathway by downregulating p-Akt and p-mTOR Inducing autophagy via activation of LC3-II Synergic therapeutic effect with Fluorouracil	Akt/mTOR MAPK
				In vivo (SAS) BALB/c nude mice, AnN.Cg-Foxn1nu/CrlNarl	Right flank subcutaneous inoculation Oral administration of 5 and 15 mg/kg dosage	Inhibiting tumor growth and inducing apoptosis	
6	Chen et al. [65]	35785707	Trichodermin	In vitro (Ca922, HSC3)	Cell Viability; MTT assay (IC_50_ value; 9.65 µM, 11.49 µM) Colony Formation; clonogenic assay (3 & 10 µM) Cell Cycle Arrest; flow cytometry (3 & 10 µM) Apoptosis; Annexin V assay; nuclear condensation observation (3 & 10 µM) Cell Migration and Invasion; transwell assay (3 & 10 µM)	Reduced cell viability, migration and invasive Downregulation of MMP-9, inhibiting cell migration and invasion Downregulation of cyclin A, cyclin D1, Cdk1/Cdk2 and Cdk4 expression, leading to G_2_/M cell cycle arrest Inducing apoptosis by upregulating Caspase-3 and cleaved PARP expression Reduces mitochondrial membrane potential, basal respiration, ATP production, maximum respiration and proton leak, inducing intrinsic apoptosis Downregulated expression of HDAC-2, phosphorylated STAT3 and NF-κB	HDAC-2 Caspase
				In vivo (HSC3) Zebrafish	Embryo tumor transplantation Embryo submerged in dosage of 3 and 10 µg/mL solution	Inhibited tumor growth	
7	Wang et al. [24]	31750253	Psorachromene	In vitro (SAS, OECM1)	Cell Viability; sulforhodamine B assay (25 & 50 µM) Cell Cycle Arrest; flow cytometry (25, 50 & 75 µM) Apoptosis; TUNEL assay (50 µM) Cell Migration and Invasion; wound healing assay and invasion assay (25, 50 & 75 µM)	Reduced cell growth and colony formation Induced cell cycle arrest in G_2_ phase Induced apoptosis via activation of Caspase-9 and cleavage PARP Inhibited cell migration and invasion, via downregulating EMT-promoting protein, Vimentin, Slug and EGFR signaling pathway Synergic therapeutic effect with Cisplatin and Doxorubicin Upregulating growth arrest specific 5 (GAS5), inhibiting cell growth and metastasis	EGFR EMT-related Caspase SLUG
				In vivo (SAS) BALB/c nude mice	Right flank subcutaneous inoculation Intraperitoneal administration of 100 µL dosage	Downregulated EMT-promoting protein and EGFR signaling pathway Reduced tumor growth	
8	Li et al. [56]	35904511	Protocatechuic acid	In vitro (HSC3, CAL27)	Cell Proliferation; CCK-8 assay (250 & 500 µM) Cell Cycle Arrest; flow cytometry (250 & 500 µM) Apoptosis; TUNEL assay (250 & 500 µM) Tumorsphere Formation; tumorsphere assay (250, 500 & 1000 µM)	Induced cell death by interrupting Serpinb9 and granzyme B (Sb9-GrB) complex formation Induced cell apoptosis via upregulated phosphorylation of c-Jun N-terminal kinase (JNK), p38 and cleaved Caspase-3 expression Upregulated superoxide dismutase and nuclear factor erythroid 2-related factor 2 (Nrf2), reducing the ROS levels Inhibiting cancer stemness	JNK/p38 Caspase
				In vivo (CAL27) Nude mice	Right flank subcutaneous inoculation Peritumoral administration of 100 µL dosage	Inhibiting tumor growth Reducing population of Ki-67 positive cell	
9	de Compos et al. [110]	28782139	Curcumin	In vitro (CAL27, SCC25, HACAT, NIH-3T3)	Cell Proliferation; NF cell proliferation assay (2, 5, 10, 20, 30, 40 & 50 µM) Cell Apoptosis; Annexin V assay (5 & 50 µM) Cell Migration and Invasion; time-lapse analysis (2 & 5 µM) Tumorsphere Formation; spheroid assay (10,20, 50 & 200 µM)	Reducing cell proliferation Reduced migratory rate and impairment on tumor cell directionality Reducing cell-cell adhesion, leading to less homogenous spheroid	
				In vivo (HNSCC Biopsy) BALB/c nude mice	Right flank subcutaneous inoculation Treatment with 70 mg/kg dosage	Inhibiting tumor growth Induced a less aggressive histological phenotype	

**Table 3 biomedicines-11-02359-t003:** Chemoprevention findings from preclinical studies with natural products on induced carcinogenesis.

No	Author	PMID	Natural Product	Preclinical Model	Test & Dosage	Findings	Pathway Related
1	Vincent-Chong et al. [25]	30875566	Calcitriol	In vivo (4NQO-induced carcinogenesis) C57BL/6NCr mice	0.1 µg dosage, intraperitoneal administration (thrice weekly)	Inhibition of 4-nitroquinoline-1-oxide (4NQO)-induced carcinogenesis Reduced incidence of HNSCC induced by 4NQO Increased Ki-67 positive dysplastic epithelium Carcinogenesis influenced by stage of intervention and duration of exposure to calcitriol	
2	Wang et al. [64]	33156559	Neferine	In vivo (DMBA-induced carcinogenesis) Syrian hamster	15 mg/kg dosage, intragastric administration (thrice weekly)	Increased body weight and suppression on formation of 7,12-Dimethylbenz[a]anthracene (DMBA) induced tumor development Downregulated expression of NF-κB, mutant p53 and PCNA	NF-κB
3	Sophia et al. [47]	30352996	Nimbolide	In vivo (DMBA-induced carcinogenesis) Syrian hamster	100 µg/kg dosage, intragastric administration	Inhibited PI3K/Akt/GSK-3β signaling pathway Sensitized tumor to apoptosis	PI3K/Akt/GSK-3β

## Data Availability

The data that support the findings of this study are available from the corresponding author upon reasonable request.

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
