# Peer review of "A Comprehensive Review of Natural Products as Therapeutic or Chemopreventive Agents against Head and Neck Squamous Cell Carcinoma Cells Using Preclinical Models"

_biomedicines, 2023, doi:10.3390/biomedicines11092359_

Round 1

Reviewer 1 Report

Thank you for inviting me to review this well executed review.

Undoubtedly, the authors made a significant effort to provide a comprehensive literature review of existing evidence concerning natural products and their potential use as therapeutic agents in management of head and neck cancer, squamous cell carcinoma type.  

Major comments:

1. Lack of critical review of in vitro study results, particularly regarding highly heterogeneous material and methods. The absence of critical comparison may lead to publication bias issue and overestimated conclusions. 

2. Lack of structured, standardised approach while conducting databases search. Some valuable articles in this field have been omitted. No reason provided why authors decided to include only studies published in the past 5 years. 

3. On the other hand, some articles from 2023 (most recent months) involving in vitro studies on SCC seem missing.    

4. Rather arbitrary selection of publications which were subject to detailed descriptive review, without a clear justification of authors' choice.  

5. A statement emphasising that the results of pre-clinical studies cannot be directly extrapolated into clinical practice must be clearly presented.  

6. The main title may need to be amended as described natural products can be potentially utilised as either therapeutic or chemopreventive agents.   

Only minor corrections required. 

Author Response

Reviewer 1

Thank you for inviting me to review this well executed review.

Undoubtedly, the authors made a significant effort to provide a comprehensive literature review of existing evidence concerning natural products and their potential use as therapeutic agents in management of head and neck cancer, squamous cell carcinoma type. 

Response: Thank you for your positive feedback on the review. I appreciate your recognition of the authors' efforts in providing a comprehensive literature review on natural products and their potential therapeutic applications in managing head and neck cancer, specifically squamous cell carcinoma type. Your acknowledgement of the well-executed nature of the review is encouraging, and it reinforces the authors' commitment to delivering a thorough and valuable contribution to the field. We are grateful for your time and expertise in reviewing this work.

Major comments:

  1. Lack of critical review of in vitro study results, particularly regarding highly heterogeneous material and methods. The absence of critical comparison may lead to publication bias issue and overestimated conclusions.

Response: Thank you so much for your comment. We acknowledge the lack of critical review of the in vitro study, which can potentially lead to publication bias issues and overestimated conclusions. We have made every effort to address the comment and have provided the necessary amendments. These amendments are indicated in Table 1, Table 2 and Table 3, spanning from page 8 to 15 and 18. Additionally, we have included a critical review of the in vitro study results, particularly focusing on the highly heterogeneous material and methods. These findings can be found at section 2.7, line 273-308, highlighted in red.

  1. Lack of structured, standardised approach while conducting databases search. Some valuable articles in this field have been omitted. No reason provided why authors decided to include only studies published in the past 5 years.

Response: Thank you so much for the information, and it is a fair comment. This comprehensive review focuses on natural products as therapeutic agents for head and neck squamous cell carcinoma (HNSCC) using preclinical models. The initiation of this review was prompted by a recent publication by Crooker et al. (PMID 29602908) that provided information on the use of natural products in HNSCC preclinical models in 2018. Over the past five years, there have been limited efforts to publish reviews related to natural products, with one notable publication by Aggarwal et al. (PMID 34354591) focusing solely on HPV-related HNSCC. To address this paucity of information, we explored the possibility of using the latest research that has not been discussed in previous review papers, particularly targeting HNSCC, as some signaling pathways are common among different subtypes. The review also highlights various challenges in the field that hinder the clinical translation of natural products. These amendments can be found at section 4, line 115-124, highlighted in red.

  1. On the other hand, some articles from 2023 (most recent months) involving in vitro studies on SCC seem missing.

Response: Thank you so much for the comment. We have included the latest published articles (PMID: 37371038 and PMID: 32859062) that involve in vitro studies on HNSCC. The amendments have been added to Table 1, specifically in entries 10 (kaempferol and fisetin) and 11 (resveratrol and curcumin). Both studies are elaborated upon in section 2.5, lines 243-244, 246, and section 2.6, line 269, highlighted in red.

  1. Rather arbitrary selection of publications which were subject to detailed descriptive review, without a clear justification of authors' choice.

Response: Thank you so much for the information, and it is a fair comment. This comprehensive review focuses on natural products as therapeutic agents for head and neck squamous cell carcinoma (HNSCC) using preclinical models. The initiation of this review was prompted by a recent publication by Crooker et al. (PMID 29602908) that provided information on the use of natural products in HNSCC preclinical models in 2018. Over the past five years, there have been limited efforts to publish reviews related to natural products, with one notable publication by Aggarwal et al. (PMID 34354591) focusing solely on HPV-related HNSCC. To address this paucity of information, we explored the possibility of using the latest research that has not been discussed in previous review papers, particularly targeting HNSCC, as some signaling pathways are common among different subtypes. The review also highlights various challenges in the field that hinder the clinical translation of natural products. These amendments can be found at section 4, line 115-124, highlighted in red.

In this case, we have incorporated the justification in this manuscript to reduce the possibility of sounding like an arbitrary selection of publications.

  1. A statement emphasising that the results of pre-clinical studies cannot be directly extrapolated into clinical practice must be clearly presented.

Response: Thank you and the statement has been incorporated in line section 4, line 137-144, highligted in red.

  1. The main title may need to be amended as described natural products can be potentially utilised as either therapeutic or chemopreventive agents.

Response : Thank you for the suggestion. The amendment to the title has been made accordingly. The new title for this manuscript is "A Comprehensive Review of Natural Products as Therapeutic or Chemopreventive Agents for Head and Neck Squamous Cell Carcinoma Using Preclinical Models."

Reviewer 2 Report

In the review the authors described natural compounds potentially suitable as adjuvant, neoadjuvant chemotherapy or chemopreventive agents. The work is well build and developed but in some point is inaccurate. The Authors speaking about the compound discussing their therapeutic potential but in many cases the compounds are studied only in animal model at the begging state and more studies on safety and efficacy are need to peaking about therapeutic potential. The authors need to modified these statements with others more appropriate to the drug studies of compounds.  

Author Response

Response: Thank you for the feedback and highlighting the need for more accurate statements regarding the therapeutic potential of natural compounds. We acknowledge that in some cases, the compounds discussed in the review are primarily studied in animal models or in early stages of research. Therefore, we understand the importance of modifying our statements to reflect the current state of knowledge and drug studies. By modifying the sentence in this way, we aim to provide a more accurate representation of the current understanding and highlight the need for additional research and clinical studies to validate the therapeutic potential of natural compounds in HNSCC treatment. The amendments were done and indicates at section, 1, line 89-93, highlighted in red.

Round 2

Reviewer 1 Report

Thank yo for your timely response and additions made.

I would suggest the essential alteration of the main title: against......carcinoma cells...   

No major concerns, however, the main text requires thorough proof-reading and some rephrasing to comply with English language standards.  

Author Response

Dear reviewer,

Thank you very much for your comment. We have rephrased our title based on your feedback and proofread the sentence to ensure compliance with English language standards.

Regards, 

Vincent
